# Monitoring of Ultra-High Dose Rate Pulsed X-ray Facilities with Radioluminescent Nitrogen-Doped Optical Fiber

**DOI:** 10.3390/s22093192

**Published:** 2022-04-21

**Authors:** Jeoffray Vidalot, Cosimo Campanella, Julien Dachicourt, Claude Marcandella, Olivier Duhamel, Adriana Morana, David Poujols, Gilles Assaillit, Marc Gaillardin, Aziz Boukenter, Youcef Ouerdane, Sylvain Girard, Philippe Paillet

**Affiliations:** 1Commissariat à l’Energie Atomique et aux Energies Alternatives (CEA), Direction des Applications Militaires (DAM-DIF), F-91297 Arpajon, France; julien.dachicourt@cea.fr (J.D.); claude.marcandella@cea.fr (C.M.); olivier.duhamel@cea.fr (O.D.); philippe.paillet@cea.fr (P.P.); 2Laboratoire Hubert Curien, Université Jean Monnet, CNRS 5516, IOGS, F-42000 Saint Etienne, France; cosimo.campanella@univ-st-etienne.fr (C.C.); adriana.morana@univ-st-etienne.fr (A.M.); aziz.boukenter@univ-st-etienne.fr (A.B.); youcef.ouerdane@univ-st-etienne.fr (Y.O.); sylvain.girard@univ-st-etienne.fr (S.G.); 3Centre d’Etudes de Gramat (CEG), Commissariat à l’Energie Atomique et aux Energies Alternatives (CEA), F-46500 Gramat, France; david.poujols@cea.fr (D.P.); gilles.assaillit@cea.fr (G.A.); marc.gaillardin@cea.fr (M.G.)

**Keywords:** radiation effects, optical materials, radioluminescence, optical fibers, X-rays, beam monitoring

## Abstract

We exploited the potential of radiation-induced emissions (RIEs) in the visible domain of a nitrogen-doped, silica-based, multimode optical fiber to monitor the very high dose rates associated with experiments at different pulsed X-ray facilities. We also tested this sensor at lower dose rates associated with steady-state X-ray irradiation machines (up to 100 keV photon energy, mean energy of 40 keV). For transient exposures, dedicated experimental campaigns were performed at ELSA (Electron et Laser, Source X et Applications) and ASTERIX facilities from CEA (Commissariat à l’Energie Atomique—France) to characterize the RIE of this fiber when exposed to X-ray pulses with durations of a few µs or ns. These facilities provide very large dose rates: in the order of MGy(SiO_2_)/s for the ELSA facility (up to 19 MeV photon energy) and GGy(SiO_2_)/s for the ASTERIX facility (up to 1 MeV). In both cases, the RIE intensities, mostly explained by the fiber radioluminescence (RIL) around 550 nm, with a contribution from Cerenkov at higher fluxes, linearly depend on the dose rates normalized to the pulse duration delivered by the facilities. By comparing these high dose rate results and those acquired under low-dose rate steady-state X-rays (only RIL was present), we showed that the RIE of this multimode optical fiber linearly depends on the dose rate over an ultra-wide dose rate range from 10^−2^ Gy(SiO_2_)/s to a few 10^9^ Gy(SiO_2_)/s and photons with energy in the range from 40 keV to 19 MeV. These results demonstrate the high potential of this class of radiation monitors for beam monitoring at very high dose rates in a very large variety of facilities as future FLASH therapy facilities.

## 1. Introduction

The development of radiation sources based on particle accelerators and radioactive sources has generated, in the last three decades, the need for different beam instrumentation and radiation sensors [1]. Their goal is to characterize the main beam parameters, such as dose-rate (or flux), temporal shape or beam size, from the production point to the device under test. Thanks to the rise in new technologies and pushed by the development of new facilities, beam diagnostics are becoming more and more performant and precise, and thus more prone to satisfy facility coordinators as well as users coming for radiation testing. For radiation testing, e.g., for electronic component radiation hardening for space or other applications [2,3], tests with pulsed beam are needed to study the influence of dose rate on device degradation basic mechanisms.

In this case, the number and the time shape of the pulses are crucial beam parameters to be monitored. In another research domain, for some therapy treatments such as FLASH [4], based on the use of X-rays to cure tumoral cells, the dose deposited in the tissue of the patient needs to be precisely quantified, even when deposited at a very high dose rate [5]. In this way, most of the radiation monitors are designed to measure the total ionizing dose (TID) deposited in the materials such as thermoluminescent dosimeters (TLD) [6] or MOSFET (Metal-Oxide Semiconductor Field Effect Transistor) detectors [7,8]. Some provide real-time monitoring of the dose rate as sensors based on scintillating materials and often also allow the TID to be deduced too [9]. All these radiation monitors are able to operate over a certain dose range (or dose rates), and generally, most of the technologies are not well adapted to high dose rates. To cover various needs, one should then combine different sensor technologies, and consequently, the amount and complexity of the beam instrumentation.

Identifying a single technology adapted to both steady-state and FLASH dosimetry is actually a very timely challenge in medicine, but also for other research domains such as fusion by inertial confinement [10]. With a steady-state radiation source, the dose rate is maintained or slowly evolves over time until the source is shut down. For pulsed sources, the dose rate has to be characterized for each pulse and usually for short-duration pulses, typically shorter than the millisecond. This adds a new constraint coming from the time shape of the beam, often very short (µs scale) or ultra-short (ns scale). If the response of the dosimeter is longer than the delay between successive pulses, the dose rate could be misevaluated. So, the pulse discrimination is also a crucial aspect for most of the targeted applications.

In this context, specialty silica-based optical fibers (OFs) with appropriate core doping are emerging as promising radiation sensors and dosimeters [11]. First, they could provide high spatial resolution thanks to their small size, and they are also immune to most of the electromagnetic perturbations considering the dielectric nature of silica. The fiber core, acting as the sensitive element, has a diameter ranging from less than a tenth to hundreds of µm. By changing its composition through the incorporation of different dopants in the silica matrix such as phosphorus (P), germanium (Ge), gadolinium (Gd), cerium (Ce), nitrogen (N), etc., the fiber radiation response can be tuned to make them more suitable to monitor dose and dose rate [12,13,14,15]. 

Fiber-based dosimeters mostly exploit two main radiation effects [16] on silica. The first is an opacification of the fiber core due to the generation of absorbing defects in the doped silica glass network [17] and is called Radiation-Induced Attenuation (RIA). This effect generally depends on the dose, the dose rate, the irradiation temperature [18], and sometimes on the irradiation type. However, it was demonstrated that the RIA at 1.55 µm of a P-doped optical fiber linearly depends on the ionizing dose (up to 300 Gy(SiO_2_) for different irradiation types (γ- and X-rays, protons, neutrons, …) and consequently, it can be used for dose monitoring in mixed environments [19,20]. Furthermore, the RIA vs. dose dependence is, for this particular class of fibers, quasi-independent on the dose rate, temperature and particle type. The second radiation-induced phenomenon is called Radiation-Induced Emission (RIE). It is composed either of Cerenkov radiation emissions [21] or of Radiation-Induced Luminescence (RIL) or by a combination of the two processes. Cerenkov radiation is initiated by the interaction between a medium and a charged particle moving faster than light in the considered medium. An electromagnetic echo comes with the charged particle with a light spectrum inversely proportional to λ^2^, with λ being the photon wavelength from the far UV to the near infrared. RIL, instead, is due to the presence of trap levels in the energy gap of SiO_2_ structure caused by pre-existing or radiation-induced centers in the silica network. When the material is irradiated, the energy transferred into the silica material is large enough to create electron–hole pairs. The expelled electrons decay into these trap levels and then recombine in the valence band or with a lower state in the silica band gap by emitting a photon. This recombination produces an RIL, whose intensity depends on the dose rate. After the calibration of the RIL as a function of dose rate, this process can be used for dose rate monitoring and, consequently, dose measurements if the irradiation duration is known [22]. In the literature, previous studies have shown the potential of waveguides manufactured using sol–gel techniques with Ce- or Gd-doped cores in the radio and proton therapy dose rate range [23,24]. Previous studies also showed that optical fibers doped with nitrogen (N) exhibited a strong RIL signature in the visible domain, adequate for the flux monitoring of X-rays or 63 MeV protons [25]. In this article, we propose to extend these previous studies by characterizing the potential of this N-doped multimode optical fiber to monitor the very high or ultra-high dose rates associated with high-energy X-ray flash facilities.

## 2. Materials and Methods

### 2.1. Facilities Description

*Microsecond scale pulsed X-ray source*—The experiments were performed at the ELSA (Electrons et Laser, Sources X et Application) facility of CEA DAM in Arpajon, France. The facility is based on an electron linear accelerator. Electron bunches are extracted by laser pulses from a photocathode placed inside a first RadioFrequency (RF) cavity, which accelerates them up to 2 MeV. The base frequency is 144 MHz, corresponding to a period of 6.9 ns. At the exit of this gun, the electrons are then guided through three RF accelerating cavities to reach energy typically adjustable between 2 and 19 MeV. 

The electron bunches are propagated towards a tantalum target to convert them into pulses of bremsstrahlung X-rays with an energy widely distributed up to the kinetic energy of the incident electrons (19 MeV, maximum), resulting in the typical energy spectrum shown in Figure 1. Pulses consist of trains of 80 ps-bunches at 144 MHz. The number of delivered bunches can be set between 1 and 500 pulses, corresponding to a duration freely adjustable from 80 ps (unique bunch) to 3.5 µs. We typically used pulse durations from a few hundreds of nanoseconds to 3 µs. The repetition rate of ELSA is 1 Hz. In this work, we only consider the deposited dose per pulse that was estimated between 1.5 and 4 Gy(SiO_2_) at 2 cm from the target. This value was obtained by multiplying the dose rate (a few MGy(SiO_2_)/s) by the bunch duration. At this facility, these two parameters were only measured for the last pulse of each irradiation run, then the deposited dose was estimated only for this last pulse. Concerning the X-ray beam shape, the 10% homogeneity area at 1 cm from the target was on a 1 cm diameter circle. The further away from the target, the larger the homogeneity area. However, the dose rate decreased with the radius perpendicular to the propagation axis. For our experiments, in order to reach the highest dose rate, we placed the sample at 2 cm from the target.
Figure 1MCNP6 (Monte-Carlo N-Particle Transport Code version 6) [26] simulation of the energy spectrum at 1 cm from the multilayer electron X-ray converter.
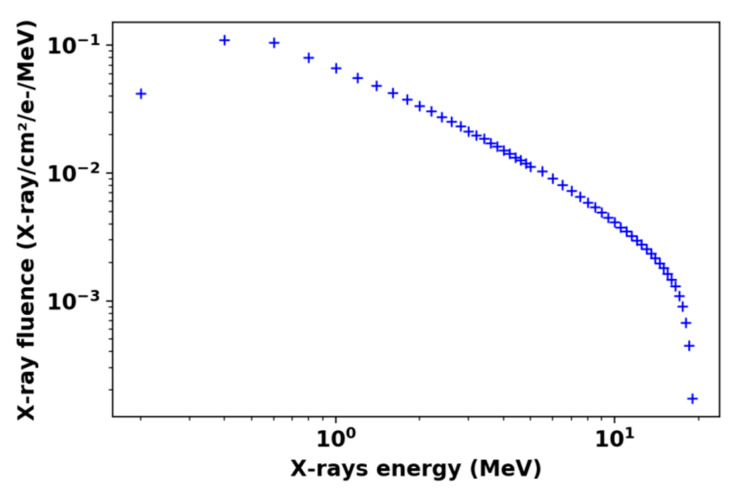


*Nanosecond scale pulsed X-ray source*—We also studied the RIE production in the N-doped sample using the ASTERIX facility at CEG, CEA, Gramat in France. ASTERIX is based on a Marx generator, delivering very high voltage discharge on a conversion photodiode. The conversion of the discharge produces an ultra-high dose rate X-ray flash with a typical temporal FWHM (Full Width at Half Maximum) as long as the discharge time of the Marx generator. The energy distribution follows an X-ray bremsstrahlung spectrum centered around 1 MeV [27,28]. Due to its mode of operation, only a single X-ray shot can be performed with this facility, with the typical delay between two shots being of about 30 min. To evaluate the dose and the dose rate delivered by the X-ray shot, facility operators use calibrated LiF TLDs. Two TLDs were located next to the fiber samples under test to measure the TID of each shot.

### 2.2. Sample Description

*Microsecond Scale Pulsed X-ray Sample*—The fiber under test was a multimode (MM) nitrogen-doped fiber (~50 µm large core) manufactured by iXblue (Lannion, France), in the framework of collaboration with Laboratoire Hubert Curien of Université Jean Monnet (St Etienne—France) and CEA [29,30]. This fiber was manufactured using the Surface Plasma Chemical Vapor Deposition (SPCVD) technique, which allows nitrogen incorporation in the fiber core [31]. This process usually leads to doped glass with a quite high level of chlorine impurity too [32]. It has been already demonstrated that N-doped glass exhibits a radioluminescence band centered around 550 nm, under X-, γ-rays and protons [25]. A 2 cm long sample of the N-doped fiber was spliced to a 1.5 m long radiation-hardened MM optical fiber (RHF, F-doped fiber, Radhard), which was connected to a light-shielded, radiation-tolerant MM optical fiber (Ge-doped) to transport the RIE to the detector, placed out of the irradiation area. This light-shielded fiber was chosen to move the acquisition system as far as possible away from the irradiation area and to reduce the parasitic ambient light as much as possible. Indeed, during the irradiation runs, even if most of the ambient lights were turned off, there was still some parasitic light coming from the safety devices within the accelerator and giving rise to a background noise signal. This noise contribution was measured and subtracted from our RIE measurements. Figure 2 illustrates the experimental setup and this irradiation facility.

*Nanosecond scale pulsed X-ray sample*—For the ns scale pulse experiment, the sample was prepared differently. Due to the ultra-high dose rate range produced by the ASTERIX facility, to obtain the correct measurements and distinguish the RIE produced by the N-doped sample rather than by the longer radiation-hardened transport fiber (that presents a low but non-null RIE signal [33]), the sample was prepared in two steps. The optical fiber transport line, identical in the µs-scale-pulsed and ns-scale-pulsed X-ray samples, consisted of the transport fiber shielded against ambient light, which linked the instrumentation to the optical fiber under radiation. Then, the first test line was only constituted by the radiation-hardened fiber to extract its RIE (luminescence and also Cerenkov radiation for this facility). The length of the RHF sample exposed to the X-ray beam was 15 cm. To join the light-shielded optical fiber, the total length of the RHF was ~2 m. Out of the 15 cm radius zone centered on the maximum dose spot, the dose deposited by the X-ray beam could be neglected. Over the 15 cm long optical fiber, 5 cm was placed in a zone where the dose was higher than 50% of its maximum value. Moreover, the optical fiber was placed perpendicular to the beam axis, allowing us to reduce the development of Cerenkov radiation as much as possible, which strongly depends on the fiber angle with the beam axis [34]. Two samples with identical lengths were prepared for RIA and RIE measurements, respectively. After the first tests on these fibers, which gave information about the effects of the dose deposited in a few ns, we could conclude that the RIA was small enough to keep the RHF as the radiation-hardened transport line. The highly irradiated 3 cm section of the RHF line was cleaved and then spliced with 10 cm of N-doped optical fiber to replace the deleted part in both samples. Placed in front of the beam, the main irradiated fiber was the N-doped one, with the extremities composed of the RHF. Finally, the setup and this irradiation facility are represented in Figure 3.

### 2.3. RIE Experimental Setup

We used a photomultiplier tube (PMT—Hamamatsu H7421 [35]) with its photon-counting unit to detect the RIE. The PMT is firstly composed of a photocathode that converts the incident RIE photons into a photoelectron current. Then, they are accelerated by an electric field towards couples of dynodes to produce electron cascades, which amplify the initial electron current. This signal is amplified to deliver a positive logical pulse shaped signal. At the PMT output, the signal is sent to a counting unit (C8855-01 [36]) which integrates the logic signal during an integration time window defined by the user. For all our experiments, we selected a time window that allowed a minimum light background to be integrated while maintaining a large dynamic. For the ELSA experiments, the integration time window was set to 100 ms, whereas 10 ms was used for the ASTERIX ones. The first integration window was chosen to keep a long period of data acquisition to measure without splitting the long set of pulses into multiple measurements. In the ASTERIX case, the acquisition was centered on the pulse production, to which we added the measurement of the background (kept as low as possible with some light shield) and the delays both before and after the shot to record the whole luminescence tail produced by the sensitive optical fiber. 

### 2.4. Experimental Procedure

*ELSA Microsecond-scale Pulsed X-rays*—For all the pulse series, the PMT was set with an integration time of 100 ms. The signal recording started a few seconds before each irradiation to acquire the background noise induced by all the parasitic lights. Moreover, the recording was stopped a few seconds after the end of irradiation to acquire the RIE afterglow (if any) of the N-doped fiber. The RIE intensity was obtained by subtracting the background level from the RIE signal. The first was determined as the average of the signal measured using the PMT before the irradiation start, whereas the second one was the average of the signal once it stabilized during the run.

*Nanosecond-scale pulsed X-ray source test*—With the ASTERIX facility, we only performed a few shots on the N-doped optical fiber. To optimize the tests, two samples were simultaneously irradiated at two different distances (31 and 45 cm) from the source, corresponding to two different dose rates (of ~8 × 10^9^ and ~5 × 10^9^ Gy(SiO_2_)/s, respectively). 

### 2.5. Calibration of the Nitrogen-Doped Optical Fiber Dosimeter

We also performed a calibration of the RIL response of the investigated N-doped optical fiber under X-rays. A 1 cm long sample was irradiated with the 40 keV X-ray machine LabHX at Laboratoire Hubert Curien (St-Etienne, France). RIL measurements (no Cerenkov in those conditions because of the low-energy X-rays) were performed by varying the dose rate from 0.03 Gy(SiO_2_)/s to 15 Gy(SiO_2_)/s. The dosimetry was achieved with an ionization chamber (PTW) placed in front of the X-ray beam at a distance from the X-ray tube window to reach the investigated dose rates by only changing the X-ray tube current. The dose homogeneity was determined by moving the dosimeter in an area as large as the sample and was almost 95% guaranteed. After the dosimetry mapping, the PTW was replaced by the N-doped sample. The irradiation setup is illustrated in Figure 4.

## 3. Results

### 3.1. LabHX Steady-State X-ray Irradiation

At each dose rate accessible with the X-ray machine operated at 100 kV, we performed a 1-minute-long irradiation run, allowing us to evaluate the RIL intensity versus dose rate. The RIL signal evolution under steady-state X-ray started from the noise background and reached a relatively flat intensity plateau during the X-ray production until the irradiator was shut down. This time evolution of the RIL signal was identical in each dose rate used, only changing by the amplitude of the RIL signal following the dose rate evolution. From each irradiation set by a defined dose rate, we extracted the RIL average and the RIL standard deviation. For each experimented dose rate, the obtained results are summarized in Figure 5.

Spectral measurements from [32] confirm that in this case, the RIE was only due to the sole 550 nm radioluminescence (RIL) that is characteristic of the N-doped fiber samples. It is important to remember that under such conditions, it was already demonstrated that the performances of the 1 cm long optical fiber probe remain unaffected by RIA at least up to doses of ~100 kGy [25]. The RIL intensity follows a linear dependence with dose rate, as expressed by Equation (1): (1)$$\mathrm{RIL}\left(\frac{\mathrm{dD}}{\mathrm{dt}}\right)=A\;\times \frac{\mathrm{dD}}{\mathrm{dt}}$$
where the parameter A is ~602 (counts/Gy(SiO_2_)) for our probe. Here, the RIL linearity of the N-doped fiber was checked for dose rates up to 15 Gy(SiO_2_)/s, which are much lower than the dose rate expected at ELSA or ASTERIX. As a reminder, for FLASH treatments, dose rates above 40 Gy/s are expected.

### 3.2. ELSA Microsecond-Scale Flash Monitoring

For each irradiation run on ELSA, we obtained an RIE peak set as shown in Figure 6. This presents eight RIE peaks related to the interaction of the nitrogen-doped optical fibers with eight ELSA X-ray flashes at a dose rate of 1.3 MGy(SiO_2_)/s. The maximum value of each peak was due to the integration in a 100 ms gate window of the RIE signal caused by the fiber interaction with the X-ray pulse. The N-doped fiber was easily able to count and detect the trains of pulses in these conditions.

The recorded signal includes both RIL and Cerenkov, since, by Compton scattering, ionized electrons can be expelled with sufficient energy to produce a Cerenkov signal into the fiber cladding and/or the core. Whereas this emission is synchronous with the pulse interaction, the emitted signal by the optical fiber material at the end of the interaction shows a fast decay. In the majority of cases, this is composed by Cerenkov radiation and N-related RIL, which are integrated into the RIE peak. When optical fiber material relaxes, a luminescence tail stays active for a delay of ~200 ms. This tail can still be composed of the photons emitted by the recombination of electrons trapped at defect sites. Additionally, it is known that deep trapping carriers [37] decaying with longer lifetimes can contribute to this tail and spread the luminescence decay time. Thanks to the long delay of 1 s between two consecutive pulses, the luminescence decays enough before the next pulse, therefore not adding extra luminescence intensity to the next RIE peak. Thanks to this property, we could separate the effect induced by each X-ray pulse interaction. 

From each irradiation pulse set, the measured RIE is summarized in Figure 7 with the corresponding pulse number in the x-axis. This is composed of two main irradiation periods colored successively in blue and orange. The first period was carried out with a mean dose rate of 1.3 MGy(SiO_2_)/s. Because of the fluctuations of the accelerator, the stability of the X-ray beam production had to be tuned sometimes. To keep the stability, some beam parameters were adjusted, such as the X-ray flash duration or the electron charge responsible of the dose rate of the X-ray pulse. Because of this change, which directly affected the dose rate delivered by the accelerator, the RIE followed the same evolution, as illustrated by the step around the 1300th pulse. After the adjustment of the beam parameters to stabilize the accelerator, the used dose rate was around 1.7 MGy(SiO_2_)/s. This experiment illustrates how the fiber monitor could complement the existing technologies, allowing us to acquire key information about all the delivered pulses by the ELSA facility. In particular, the small-size optical fiber allowed us to follow the pulse-per-pulse dispersion in terms of dose rate in real-time, leading to a more precise estimation of the deposited dose on the device under test.

It is interesting to note that, for such irradiation conditions, the TID deposited on the 1 cm long probe could be quite large. Indeed, if we consider the maximum estimated TID of 4 Gy per pulse and around 5500 pulses, doses in the order of 22 kGy(SiO_2_) could be deposited. If the N-doped fiber could easily withstand such doses [25] without any impact of RIA or any degradation of its optical properties, this will probably not be the case for plastic-based scintillating fibers that are not designed to resist to doses exceeding kGy dose levels. In the future, TID tests up to MGy dose levels of the N-doped fiber probe will have to be performed to detect the maximum dose, in terms of number of pulses, which can be tolerated in function of its profile of use (length…).

Each RIE level appears related to the mean dose rate used with a dispersion of ~100 counts/100 ms. Using these results, we can start to construct a correlation between RIE produced by both steady-state X-rays and produced by µs-scale pulsed beam. 

### 3.3. Asterix Nanosecond Scale Flash Monitoring

In total, five different X-ray shots were performed on the probe samples. Before that, we checked on dedicated samples that the room-temperature (RT) stable radiation-induced attenuation (RIA) caused by X-ray shots on both the RHF transport line and the N-doped sample would not affect the RIE (RIL and Cerenkov) responses during consecutive shots. Under such high-dose-rate transient exposures, the transient RIA (during and shorter times after the pulse) is known to be very high during such an X-ray shot [25,38]. Indeed, the transient losses are very high in optical fibers because of the contribution of both RT unstable, metastable and stable defects, which bleach quickly after the pulse through thermal or photo-bleaching processes [18,38]. The impact of transient RIA on the RIE could not be avoided even though the use of small length samples reduced its impacts. After 450 s, no RIA contribution was discernible in our sensing line, ensuring that permanent RIA does not affect the measurements presented below.

Figure 8 illustrates the temporal evolution of the RIE before, during and after an X-ray shot. The RIE associated with the X-ray pulse was easily measured as well as the long-time decay of the RIL post-pulse. In Figure 8 inset, we plot the RIE intensities measured for the five X-ray pulses in function of the dose rates provided by the facility operators from their TLD dosimeters. This last figure was built using three main assumptions: The first one is that the RIE intensity measured with a gate of 100 ms is equivalent to the RIE intensity measured at 10 ms multiplied by a factor 10. This is correct as long as the detector is not operated in saturation conditions. This was verified for all the reported experiments. The second one is that the RIE intensity measured with different sample lengths could be normalized to a 1 cm long sample. This assumption leads to a certain uncertainty as RIA affects the RIE measurements in a quite complex way. Finally, the third hypothesis is that the measured RIE is produced mainly during the ns pulse interaction. With this very realistic assumption, we can multiply the measured RIE by the inverse of the temporal FWHM of the X-ray pulse. 

## 4. Discussion

Our set of experimental results showed that the RIE of the N-doped optical fiber linearly depends on the dose rate for the three facilities. In this section, we compare the sensitivities measured for the different N-doped luminescent probes at the different facilities in order to better understand if the response of the dosimeter is affected noticeably by various possible parasitic effects such as RIA. To illustrate this comparison, in Figure 9, we plot all the RIE results acquired at LabHX, ELSA and ASTERIX with different probes and at very different dose rates. To compare those results, we have to consider the equivalent RIE that should have been measured at the different dose rate considering identical duration time. We chose the integration time of 100 ms, similar to the integration time of our PMT for the ELSA and LabHX runs. Then, we could extrapolate the number of counts that will have been attained for such irradiation times at the different dose rates. As an example, by multiplying the ELSA-measured RIE values by 2.08 × 10^4^ (ratio between the 100 ms considered irradiation time versus the real 4.8 µs ELSA pulse duration), we can compare the RIE versus dose rate dependencies for the two facilities. The agreement is quite good regarding our use of samples of different lengths at the two facilities as well as the uncertainties on the last pulse duration and dose rate provided by ELSA operators. A linear fit was applied to link the two dose rate ranges. Under these assumptions, the RIE measured with the ultra-high dose rate agrees well with to the steady-state and µs-scale data. Similarly, we could analyze the ASTERIX results following a similar approach. In direct comparison with the µs-scale data, the high dose, higher than 25 Gy(SiO_2_), deposited in an ultra-short time (a few ns) at ASTERIX, generate RIE in very good agreement with those associated with the µs-scale pulse associated with lower TID of a few Gy(SiO_2_) or the steady-state LabHX measurements. A very interesting result is that we observe no noticeable overgeneration of luminescence due to the accumulated dose, known as the Bright Burn Effect [37], in our test conditions. The linear fits for these new sets of data gives a parameter A equal to 663 counts/Gy(SiO_2_) when combining steady-state X-ray and µs-scale X-ray beam interaction and 558 counts/Gy(SiO_2_) for all concatenated data. Our study demonstrates the potential of this optical fiber to monitor the photon flux over more than 9 decades of dose rate, and for photons with energy ranging from 40 keV to 19 MeV. 

All the fit parameters A used to interpolate the data are summarized in Table 1.

## 5. Conclusions

The dosimetry of irradiation facilities is becoming crucial to guarantee the reliability and the control of beams. From the operator to the user viewpoints, working with a performant and accurate dosimetry during irradiation brings confidence in the irradiation parameters and allows precise studies. In this context, silica-based optical fibers have shown high potential for dose and dose rate monitoring. In this article, we investigated the potential of a radioluminescent nitrogen-doped multimode optical fiber to monitor the dose rates associated with high-energy µs (up to 19 MeV) and ns scale (~1 MeV) X-ray pulses. This kind of optical fiber was shown to provide a linear dose rate dependence of the RIE over 9 decades of dose rate from 0.01 Gy(SiO_2_)/s (40 keV X-rays) to GGy(SiO_2_)/s according to the reduction of the pulse duration to few ns. For steady-state irradiation, µs-scale and ns-scale pulses with photons of energy ranging from tens of keV to MeV, the physical effects linking the dose rate to the radiation induced emission seemed unchanged, paving the way for dosimeters to be able to operate in an extremely large range of dose rates. The potential of such fiber-based monitors for FLASH therapies will be investigated in the future.

## Figures and Tables

**Figure 2 sensors-22-03192-f002:**
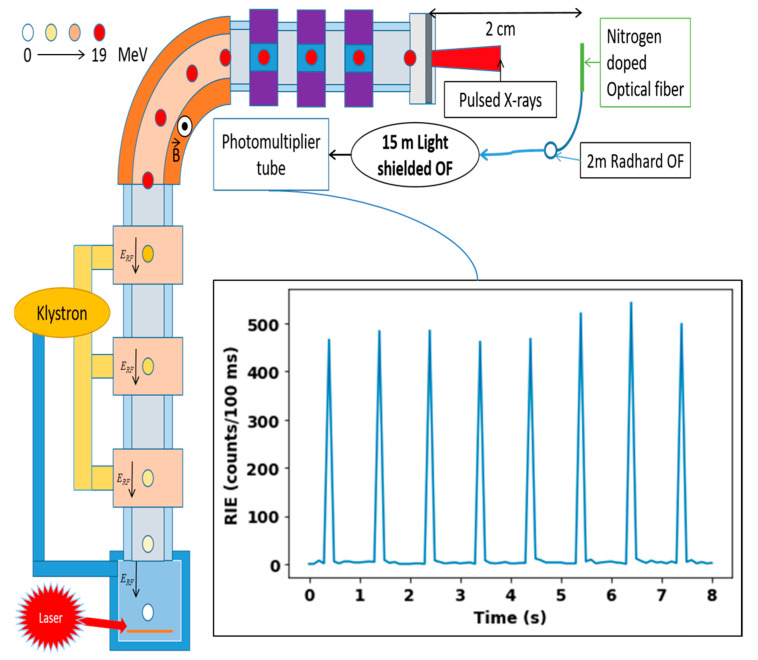
ELSA experimental irradiation area and experimental setup developed to measure the RIE induced in the N-doped fiber under test. The inset describes the temporal evolution of the RIE signal for 8 successive ELSA X-ray pulses.

**Figure 3 sensors-22-03192-f003:**
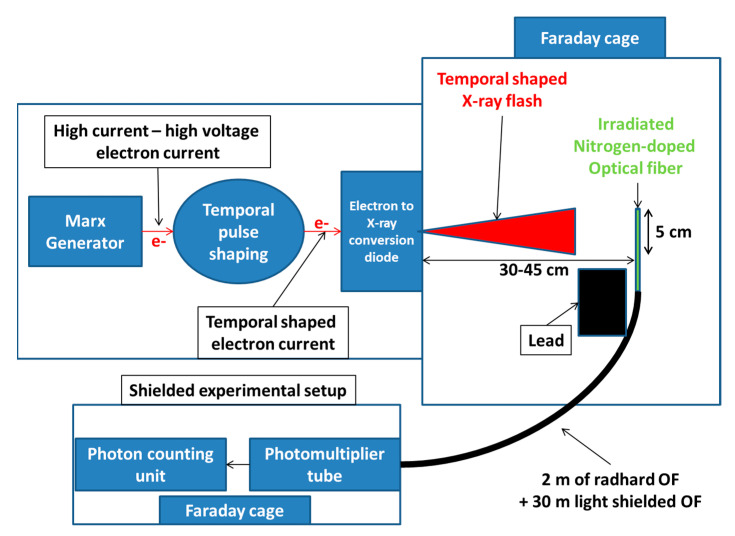
ASTERIX experimental irradiation area and experimental setup developed to measure the RIE induced in the N-doped fiber under test.

**Figure 4 sensors-22-03192-f004:**
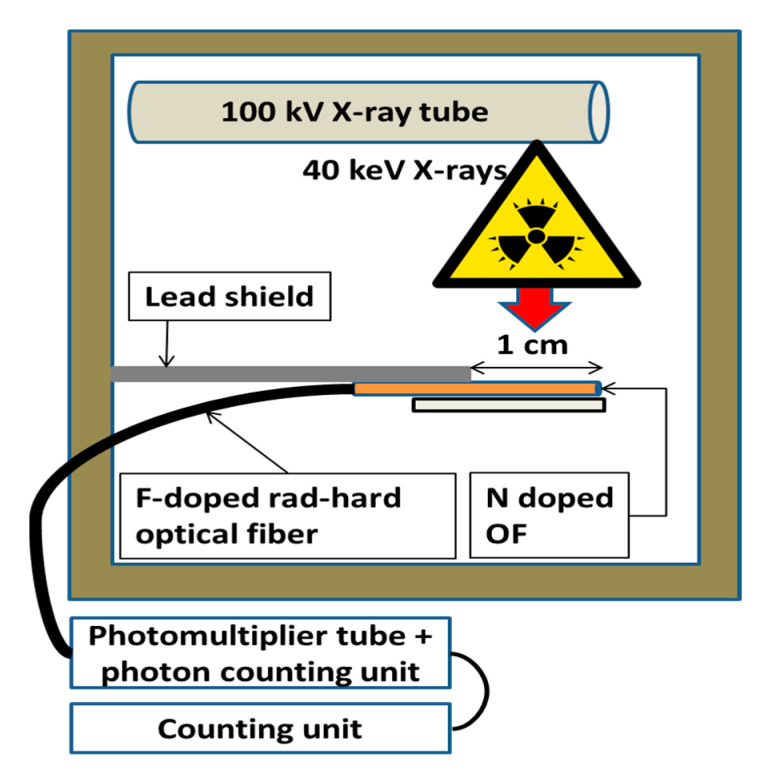
RIL versus dose rate calibration setup under steady-state X-ray irradiation.

**Figure 5 sensors-22-03192-f005:**
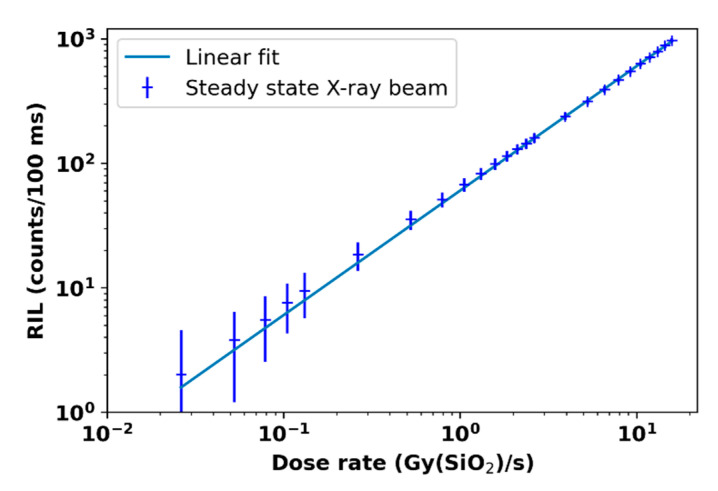
RIL dose rate dependence of the nitrogen-doped optical fiber performed with LabHX 40 keV X-ray machine.

**Figure 6 sensors-22-03192-f006:**
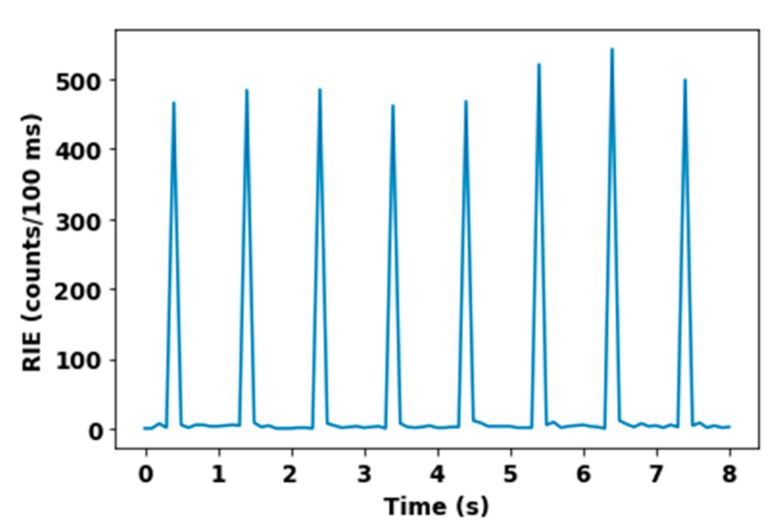
RIE peaks related to the interaction of the N-doped fiber with 8 ELSA X-ray pulses.

**Figure 7 sensors-22-03192-f007:**
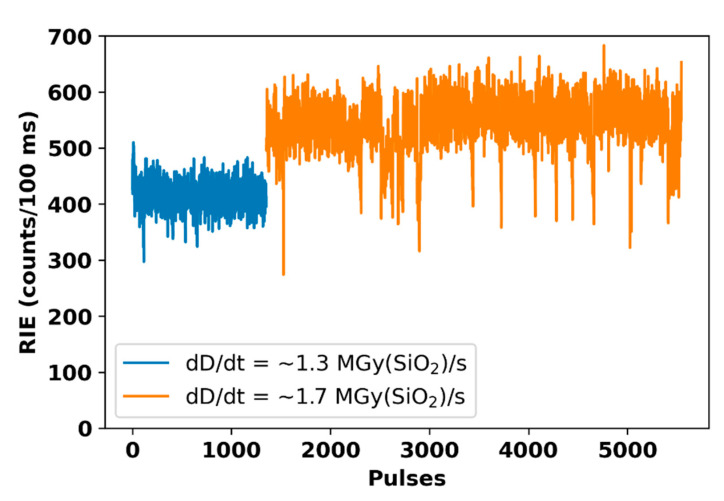
RIE evolution following the dose variations, as a function of the X-ray pulses. The first pulse set in blue was performed with a dose rate average of ~1.3 MGy(SiO_2_)/s, while the second set in pink was performed at ~1.7 MGy(SiO_2_)/s.

**Figure 8 sensors-22-03192-f008:**
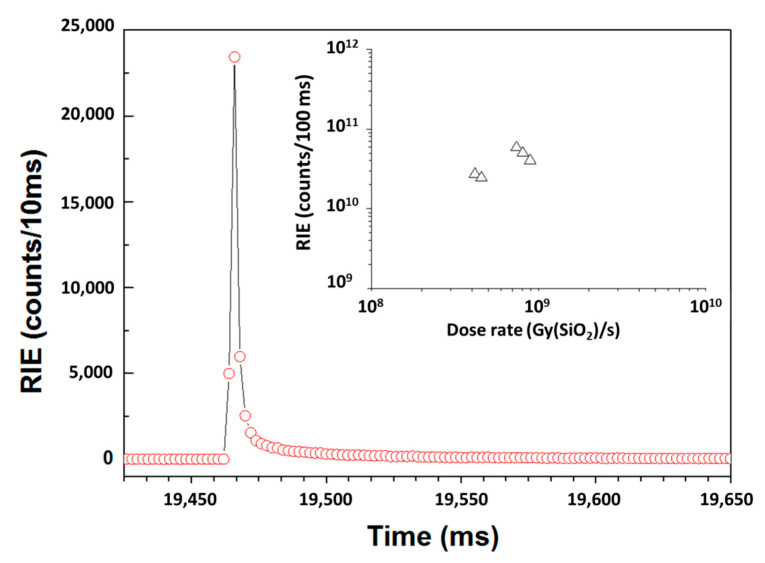
Illustration of the detected RIE for the N-doped sample before, during and after an X-ray shot at ASTERIX. In the inset are the reported normalized RIE corresponding to the dose rates associated with the 5 different shots (black triangles).

**Figure 9 sensors-22-03192-f009:**
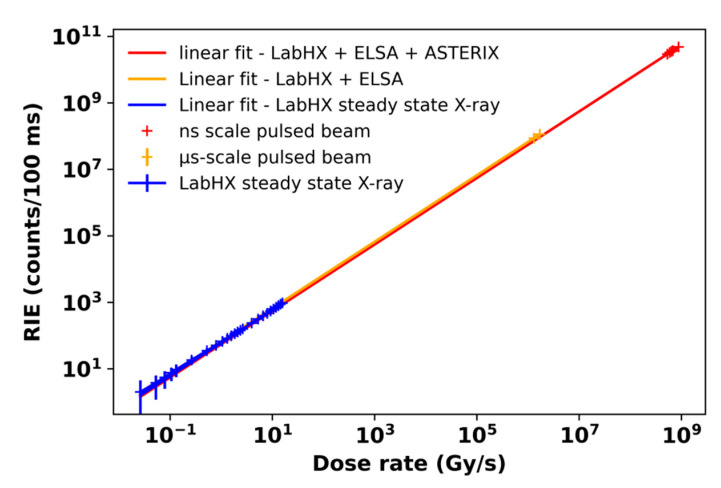
Comparison between the nitrogen-doped optical fiber RIE dose rate dependencies as measured with steady-state 40 keV X-ray source and µs (up to 19 MeV) and ns (~1 MeV) scale pulsed X-ray sources.

**Table 1 sensors-22-03192-t001:** Fit parameter used to interpolate RIL measurements.

Irradiation Type	A (Counts/Gy)
Steady-state X-ray	602
Steady-state X-ray + µs scale X-ray pulse	663
Steady-state X-ray + µs scale X-ray pulse + ns scale X-ray pulse	558

## Data Availability

Data are available upon reasonable request from the authors.

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
