# Peer review of "Monitoring of Ultra-High Dose Rate Pulsed X-ray Facilities with Radioluminescent Nitrogen-Doped Optical Fiber"

_sensors, 2022, doi:10.3390/s22093192_

Round 1

Reviewer 1 Report

This paper explored a radio luminescent Nitrogen-doped multimode optical fiber to monitor the dose rates associated with high energy μs (up to 19 MeV) and ns scale (~1 MeV) X-ray pulses. The optical fiber provides a linear dose rate dependence of the RIE over 9 decades of dose rate from 0.1 Gy(SiO2)/s (40 keV X-rays) to GGy(SiO2)/s according to the reduction of the pulse duration to few ns. Steady state irra-380 diation, μs scale and ns scale pulses with photons of energy ranging from tens of keV to MeV, the physical effects linking the dose rate to the radiation induced emission seem unchanged opening the way to dosimeters able to operate in an extremely large range of dose rates. However, some details of the discussion are needed to be specified with major revise. Out of these reasons, I can only agree to a publication of this manuscript in the sensors once these problems below have been properly addressed.

1) Why does nitrogen-doped fiber is of the characteristics of radiation characteristic measurement, dose and dose rate monitoring? Can its mechanism be further explained? 
2) The pulsed X-ray source and steady state X-ray irradiation effect on the N-doped silica fiber is different. mechanism of action on optical fiber radiation, 
3) In Fig3, where does RIE peaks related to the interaction of the N-doped fiber come from. Why is the unit counts/100ms.
4) The nitrogen-doped fiber is of the characteristics of radiation characteristic measurement. Could it be compared with plastic-based scintillating fibers, and what are its advantages.
5) After different radiation doses and dose rate treatment, how to change does the loss spectrum of the nitrogen-doped fiber. why the radiation-induced emission is in the visible domain.

Author Response

Dear reviewer,

please find our answer to your report in the following document.

Best regards.

J. Vidalot for all co-authors.

Reviewer 2 Report

The comments about the article are listed below:

-Make sure that all the acronyms are defined and are defined  the first time they appear in the manuscript 

-For the experimental setup, it should be convenient to add a scheme for illustrating the experimental setup

Author Response

(The authors gave the same response as above.)

Reviewer 3 Report

The article present the concept of using radiation induced light produced in nitrogen-doped optical fibers to monitor radiation over a wide range of dose densities. Measurements in three different irradiation regimes, from low dose continuous to very high dose ns pulsed mode, are presented. The authors show that the measured light output scales linearly over 9 orders of dose rates.

The article is well written, with understandable results and conclusions. A few minor revisions are necessary to better clarify some key points of the experiments:

(minor)

(1) The configurations, lengths and positions in the radiation filed of three different fibers used are not easy to understand from current text, especially for the Nanosecond scale pulsed X-ray sample (line 172 onwards), and the fiber modification description around line 185. The article would greatly benefit from graphical representations of all different configurations.

(2) line 188: it is not clear what the photodetector measured – the text says voltage was integrated, while the results (Figs 2-5) are expressed as Counts/time. Were individual photons counted (at what threshold)? What would be the smallest light output measurable, and what would that mean for smallest dose for which the method is usable?

(comments)

It is not clear to me, why immediate radiation-induced absorption (RIA) does not affect the linearity of measured radiation-induced emission (RIE). I would expect that at higher doses RIA would absorb relatively larger fraction of photons, than is relative increase of them due to RLE.

(language)

- line 45: test →tests ?

- line 308: Sentence not clear, did you mean: “because of the contribution of both RT unstable, metastable and stable defects bleach quickly” →“because of the contribution of both RT unstable, metastable and stable defects, which bleach quickly”

- line 380: Sentence not clear, did you mean: “Steady state irradiation...” →”For steady state irradiation...” ?

Author Response

(The authors gave the same response as above.)

Round 2

Reviewer 1 Report

The authors reply the comments well and the manuscript is suggested to be accepted.